# Simple In-House Fabrication of Microwells for Generating Uniform Hepatic Multicellular Cancer Aggregates and Discovering Novel Therapeutics

**DOI:** 10.3390/ma12203308

**Published:** 2019-10-11

**Authors:** Chiao-Yi Chiu, Ying-Chi Chen, Kuang-Wei Wu, Wen-Chien Hsu, Hong-Ping Lin, Hsien-Chang Chang, Yung-Chun Lee, Yang-Kao Wang, Ting-Yuan Tu

**Affiliations:** 1Department of Biomedical Engineering, National Cheng Kung University, Tainan 70101, Taiwanyingtzai@gmail.com (Y.-C.C.); stephen0001345@gmail.com (K.-W.W.); d2532875@gmail.com (W.-C.H.);; 2Department of Chemistry, National Cheng Kung University, Tainan 70101, Taiwan; hplin@mail.ncku.edu.tw; 3Medical Device Innovation Center, National Cheng Kung University, Tainan 70101, Taiwan; 4Center for Micro/Nano Technology Research, National Cheng Kung University, Tainan 70101, Taiwan; 5Department of Mechanical Engineering, National Cheng Kung University, Tainan 70101, Taiwan; yunglee@mail.ncku.edu.tw; 6Department of Cell Biology and Anatomy, College of Medicine, National Cheng Kung University, Tainan 70101, Taiwan; humwang@mail.ncku.edu.tw

**Keywords:** multicellular cancer aggregates, multicellular tumor spheroids, microwell, liver cancer, doxorubicin, photothermal treatment

## Abstract

Three-dimensional (3D) cell culture models have become powerful tools because they better simulate the in vivo pathophysiological microenvironment than traditional two-dimensional (2D) monolayer cultures. Tumor cells cultured in a 3D system as multicellular cancer aggregates (MCAs) recapitulate several critical in vivo characteristics that enable the study of biological functions and drug discovery. The microwell, in particular, has emerged as a revolutionary technology in the generation of MCAs as it provides geometrically defined microstructures for culturing size-controlled MCAs amenable for various downstream functional assays. This paper presents a simple and economical microwell fabrication methodology that can be conveniently incorporated into a conventional laboratory setting and used for the discovery of therapeutic interventions for liver cancer. The microwells were 400–700 µm in diameter, and hepatic MCAs (Huh-7 cells) were cultured in them for up to 5 days, over which time they grew to 250–520 µm with good viability and shape. The integrability of the microwell fabrication with a high-throughput workflow was demonstrated using a standard 96-well plate for proof-of-concept drug screening. The IC_50_ of doxorubicin was determined to be 9.3 µM under 2D conditions and 42.8 µM under 3D conditions. The application of photothermal treatment was demonstrated by optimizing concanavalin A-FITC conjugated silica-carbon hollow spheres (SCHSs) at a concentration of 500:200 µg/mL after a 2 h incubation to best bind with MCAs. Based on this concentration, which was appropriate for further photothermal treatment, the relative cell viability was assessed through exposure to a 3 W/cm^2^ near-infrared laser for 20 min. The relative fluorescence intensity showed an eight-fold reduction in cell viability, confirming the feasibility of using photothermal treatment as a potential therapeutic intervention. The proposed microwell integration is envisioned to serve as a simple in-house technique for the generation of MCAs useful for discovering therapeutic modalities for liver cancer treatment.

## 1. Introduction

Liver cancer is the second most common cause of mortality from cancer worldwide and is estimated to have been responsible for nearly 781,000 deaths in 2018 [1]. Hepatocellular carcinoma (HCC) is the most common type of primary liver cancer and is associated with poor prognosis due to the lack of early diagnosis, and treatments are yet to be fully developed [2]. Although a number of therapeutic management strategies are available, including liver transplantation, liver resection and chemoembolization, the treatment of HCC remains unsatisfactory due to the low efficacy of conventional strategies [3]. In addition, the pace of drug development for HCC has remained incredibly slow, underscoring the need for the development of new anticancer agents and therapeutic interventions [4].

The challenges of the lengthy cancer drug development process may be attributed, in part, to the lack of representative models [5]. Historically, most drug screening programs have been routinely carried out in two-dimensional (2D) cell culture because it facilitates control of the working environment and ease of handling, rendering it the most common in vitro test platform in the fields of cancer biology and medicine. However, a key limitation of 2D culture is that it fails to adequately mimic key features of the natural three-dimensional (3D) environment [6]. In the past decade, multicellular tumor spheroids (MCTSs) or multicellular cancer aggregates (MCAs) have surfaced as one of the most commonly used models for recapitulating several crucial elements of the tumor microenvironment [7]. In part, MCAs constitute a highly complex 3D arrangement of cells that can mimic enhanced cell–cell contacts, extracellular matrix (ECM) secretion, nutrients, gas exchange, structure and signaling of early-stage avascular tumors [8]. In fact, several studies observed that MCAs provide a relatively more pathophysiologically relevant environment to study biological functions. For instance, assessing the effects of drugs in 2D or 3D systems can lead to different results depending on the type of assay used, where in vitro 3D MCA models yielded more clinically relevant data than were obtained with 2D monolayer cultures [9,10].

Various approaches have been proposed to generate MCAs, e.g., nonadherent plates [11], spinner culture [12], hanging drop [13] and microfabrication techniques [14,15]. The performance criteria generally depend on trade-offs in uniformity, throughput, cost and ease of usability in producing MCAs, as well as the integration of subsequent applications (reviewed by Ahn et al. [16].) To meet the criteria, microwells produced by microfabrication have emerged as revolutionary tools over the past decade as they provide structurally confined microstructures for highly consistent and reproducible growth of MCAs (reviewed by Lee et al. [17].) Microwell fabrications are often criticized because the inherent lithography/cleanroom processes needed to produce them are expensive and the process is time-consuming. Commercial products, e.g., those from StemCell Technologies (Vancouver, Canada) and Elplasia (Okayama, Japan), provide alternative solutions by saving users from the tedious production of microwells, thereby enabling them to concentrate on biological and medical utilization of the microwells. However, commercially available products also face limitations, not only because of their costly nature, but also because these microwells are produced in fixed settings, such that they lack the flexibility for changing dimensions or arrangements. Moreover, the ability to integrate microwells with a standard microtiter platform for higher throughput and parallel data acquisition is available in few products. These difficulties currently limit microwell development, underscoring the need for a technique that enables scientists to aptly fabricate inexpensive microwells on-demand and in-house. In particular, the proposed approach demonstrates the ease of downstream workflow integration in conventional laboratory settings.

We have previously reported a rapid microwell prototyping method based on conventional petri dishes for the convenient generation of size-controlled lung MCAs [18]. In this study, we further demonstrate that the method can be integrated into a simple in-house microwell workflow through the formation of uniform 3D hepatic MCAs for exploring therapeutic interventions (Figure 1). To confirm the ease of microwell fabrication, size-controlled microwells were fabricated in either a petri dish or a microtiter plate by tuning the laser power and focusing parameters. Different parametric microwells were shown and verified to form uniformed hepatic MCAs. The identification of novel therapeutic interventions was performed by integrating microwells with either drug screening workflows or photothermal treatments. The anticancer drug doxorubicin (DOX) was administered to a microwell-modified 96-well plate for the proof-of-concept of high-throughput capability. Photothermal treatment via binding of concanavalin A (ConA)-modified silica–carbon hollow spheres (SCHSs) onto MCAs was found to yield a therapeutic effect.

## 2. Materials and Methods 

### 2.1. Microwells

The fabrication of microwells was previously described [18]. To begin with, a 25 W CO_2_ laser engraver (LaserPro Mercury II; GCC Inc., New Taipei City, Taiwan) was used to perform the microwell fabrication. It was primarily composed of a scanning head controlled by a servomotor to ensure laser ablation at the desired location in high speed. Given that the 10.6 μm CO_2_ laser can be absorbed by polystyrene (PS), resulting in regional thermal heating that makes the material melt and vaporize, single-point ablation was applied to create a concave microstructure (Appendix A). The laser was focused through a 2’’ focusing lens with varied focusing planes on the surface of a conventional untreated microtiter 96- or 12-well plate, or 60 mm petri dish (Appendix A) (Nunclon; Thermo Fisher Scientific, Waltham, MA, USA). Using a staggered arrangement for microwell arrays could save surface space, which would accommodate more microwells in a defined area (Appendix A). The optimal distance between microwells was determined according to their size. The top and side views of the concave microwells, as well as the recast region due to laser ablation, are illustrated in Appendix A.

### 2.2. Cell Culture

The Huh-7 hepatocarcinoma cell line was purchased from the JCRB cell bank (JCRB0403), cultured in 60 mm dishes (Nunclon; Thermo Fisher Scientific, Waltham, MA, USA) and maintained in high glucose Dulbecco’s modified Eagle medium supplemented with 10% fetal bovine serum (Gibco, Gaithersburg, MD, USA). The cells were grown in the presence of 5% CO_2_ at 37 °C in a humidified incubator. The medium was monitored daily and replaced with fresh medium two or three times per week. Cells that reached 80%–90% confluency were detached from the culture dish by treatment with 0.05% trypsin–EDTA (Corning, NY, USA) at 37 °C in an incubator for 3 min. Then, the culture medium was added to inhibit the enzymatic reaction. Cells were counted and assessed using an inverted microscope (Olympus, Tokyo, Japan) to ensure that an appropriate cell concentration was used during the experiments.

### 2.3. Formation of MCAs

The fabricated microwells could be stored on the shelf for up to three months and washed twice with 75% ethanol to remove debris from the laser fabrication and for disinfection prior to use. Before cell seeding, the microwells were immersed in 0.2% pluronic (F127, Sigma–Aldrich, St Louis, MO, USA) for 30 min to prevent undesired cell attachment to the microwell plate, followed by two phosphate-buffered saline (PBS; Invitrogen, Carlsbad, CA, USA) washes. Then, a cell concentration of 1 × 10^5^ cells/mL was determined based on the surface area of the culture plasticwares and seeded in the microwell-modified 60 mm petri dishes (5 mL of cell suspension media/dish), and 0.2 × 10^5^ cells/mL and 0.15 × 10^5^ cells/mL for 12- and 96-well plates (1 mL and 200 µL of cell suspension media/well), respectively. In the petri dishes and 12-well plate, excessive cells that did not lodge into the microwells were removed through medium exchange at the edge of the cell culture surface after 10 min of cells seeding at room temperature [19]. In the 96-well plate, due to the limited surface area difficult for exchanging the medium without unaffecting the cells, only half of the medium was replaced daily. The formation of MCAs was achieved by incubating cells under the same cell culture conditions for 4 days. The MCAs were retrieved by pipetting the culture medium several times to flush the MCAs out of the microwells.

### 2.4. MCA Morphology and Viability Assessment

The morphology of the MCAs generated in microwells was assessed by first removing the culture media. Then, green 5-chloromethyl fluorescein diacetate (CMFDA, 1:3000 dilution; Invitrogen, Carlsbad, CA, USA) staining solution was added directly and incubated at 5% CO_2_ and 37 °C for another 40 min, after which it was replaced by Hoechst 33258 nucleic acid staining solution (1:2000 dilution; Invitrogen, Carlsbad, CA, USA) diluted in PBS. The cells were subsequently incubated at 5% CO_2_ and 37 °C for another 40 min. Finally, the solution was removed, and 1000 µL of PBS was added to the microwells. For ease of imaging, the MCAs were transferred to a 96-well plate at a volume of 200 µL/well in suspension, and their morphology was evaluated using an inverted fluorescence microscope immediately after the MCAs settled to the bottom of the plate. Photothermal treatment on MCAs were assessed using the LIVE/DEAD viability/cytotoxicity assay kit (Invitrogen, Carlsbad, CA, USA) to analyze cell viability. Calcein-AM staining solution (1:2000 dilution) and EthD-1 staining solution (1:1000 dilution) were added and incubated at 5% CO_2_ and 37 °C for 1 h prior to image capture and examination. It should be noted that to avoid breakage or dislodging of cells from MCAs during the pipetting steps, all MCAs were kept in the microwells throughout the entire staining procedures, and only retrieved prior to the optical and fluorescent investigation.

### 2.5. Scanning Electron Microscopy

The microwell specimens were cut to the proper size, and the samples were washed several times with deionized water and ethanol. After the samples were air-dried, double-sided copper tape was used to attach them to a copper pad, where they were treated with vacuum sputtered platinum (JFC-1600; JEOL Ltd., Tokyo, Japan). The surface structures of both the microwells and MCAs were investigated by scanning electron microscopy (SEM; JSM-7001F; JEOL Ltd., Tokyo, Japan). Before the MCAs were viewed by SEM, the medium was aspirated, and the sample was rinsed with PBS. The MCAs were fixed in 0.1 M sodium cacodylate buffer (SCB; Sigma–Aldrich, St Louis, MO, USA), 1.6% PFA and 2.5% glutaraldehyde (Sigma–Aldrich, St Louis, MO, USA) for 4 h at room temperature and then rinsed with 0.1 M SCB on a shaker table for 10 min. The cells were post-fixed in 1.0% aqueous osmium tetroxide (Sigma–Aldrich, St Louis, MO, USA) in 0.1 M SCB in a dark fume hood for 90 min and then rinsed with 0.1 M SCB and placed on a shaker table for another 10 min. Finally, the samples were serially dehydrated with different concentrations of ethanol (37%, 67%, 95% and 100%) on a shaker table for 10 min each and subjected to critical point drying (CPD 030; Balzers, Liechtenstein, Germany) and vacuum sputtered platinum (JFC-1600; JEOL Ltd., Tokyo, Japan).

### 2.6. 2D and 3D Drug Screening

Both treated and untreated conventional cell cultures in 96-well plates were utilized for proof-of-concept high-throughput 2D and 3D drug screening (Appendix A). For the 2D condition, Huh-7 cells were cultured in a treated 96-well plate (0.15 × 10^5^ cells/mL). For the 3D condition, the microwells were fabricated in an untreated 96-well plate with different laser parameters. DOX stock solutions were diluted in growth medium to final concentrations of 0, 0.1, 0.32, 1, 3.2, 1, 32, 100 and 320 µM. On day 4, 200 µL of diluted DOX solution was carefully replaced with the 2D and MCA cultures and incubated for 12 h. On day 5, the supernatant was removed from the MCAs cultured in the 96-well-based microwells, and 20 µL/well CellTiter-Blue^®^ cell viability assay solution (Promega) was added to the cells to a final volume of 100 µL to assess cell viability after treatment with DOX. Then, the solutions from both 2D and 3D conditions were transferred to an opaque 96-well plate to minimize background luminescence. The different drug concentrations were made in quadruplicate, and the negative control (cell free) was used to determine any background luminescence. A luminescence reader (Luminoskan Ascent; Thermo Fisher Scientific, Waltham, MA, USA) was used to obtain luminescence intensity readouts (excitation/emission, 560 nm/590 nm). Because the luminescence intensity of CellTiter-Blue is directly proportional to cell viability, the average cell viability for each drug concentration was calculated by normalizing each value to the untreated control at 0 μM.

### 2.7. Synthesis of SCHSs

A detailed synthesis protocol for generating SCHSs was previously described [20]. In short, a stable gelatin-modified poly (methyl methacrylate) (PMMA) bead colloid mixture (gelatin-PMMA, approximately 300 nm in 25 mL) was prepared at the desired concentration under stirring for 1 h. The gelatin-PMMA bead solution was then mixed with an acidified silicate solution (a mixture of a 20 g of 3.0 wt% sodium silicate solution and 20 g of 0.15 M H_2_SO_4_) at pH 4.0. After being stirred for a few hours, a silica–gelatin–PMMA bead gel solution was formed, and it was hydrothermally treated at 100 °C for 1 day. The application of additional filtration and drying procedures resulted in silica-PMMA beads. To produce the SCHSs, the silica-PMMA beads were pyrolyzed at 800 °C for 1 h under a helium atmosphere.

### 2.8. Preparation of ConA(-FITC)-SCHSs

Either ConA- or FITC-labeled ConA (ConA–FITC) (Sigma–Aldrich, St Louis, MO, USA) at different concentrations was mixed with the SCHS solution at room temperature for 12 h with rotation. The ConA(–FITC)–SCHS complexes were then obtained by centrifugation and washed three times with PBS prior to use.

### 2.9. Photothermal Treatment

The MCAs were formed after 4 days in a 12-well plate fabricated with a microwell array (Appendix A). The ConA(-FITC) solution was centrifuged at 1800 rpm for 5 min to remove the free ConA(-FITC), and then the ConA was washed twice with PBS and resuspended in 1 mL of growth medium. The concentration and time required for binding of the ConA(-FITC) with SCHSs were experimentally determined at ratios of 25:10, 125:50, 250:100 and 500:200 μg/mL, respectively. The MCAs were incubated with ConA(-FITC)-SCHSs at 5% CO_2_ and 37 °C for 0.5 to 2.5 h in 30 min intervals prior to NIR laser illumination. The Huh-7 cells were illuminated with an 808 nm NIR laser at a power density of 0.75 W/cm^2^ from 5 to 20 min.

### 2.10. Statistical Analysis

The means and standard deviations were calculated by Student’s t-tests for multiple comparisons (n ≥ 3). All data are represented as the means ± SEM (standard error of the mean) for each experiment. Student’s t-tests were used for statistical analyses, with a *p* < 0.05 considered significant (* *p* < 0.05; ** *p* < 0.01; *** *p* < 0.001).

## 3. Results

### 3.1. Fabrication of Microwells

We fabricated microwells using an in/out-of-focus laser beam or beams of different laser power to generate microwells in a range of sizes (Figure 2). The culture plate substrate of PS was positioned for defocusing at either 0 mm, −3 mm or −6 mm below the reference focal plane, and the material was ablated at different levels of laser power: 5, 10, 15 and 25 W. The microwells that were fabricated at the focal plane (0 mm) with 5 W and 10 W were denoted M_0@5_ and M_0@10_, respectively. For the defocusing parameters at −3 mm and −6 mm, the microwells were denoted M_−3@15_ and M_−6@25_ for a laser power of 15 W and 25 W, respectively. Microwell size was incrementally adjusted by both changing the power and the location on the z-axis (Figure 2A). The SEM images of the top and side of the microwells, which had been fabricated to different depths and widths by varying the laser power and adjusting the focusing plane, showed a very smooth surface profile for all the parameters used (Figure 2B). Microwell diameter and depth were measured under different ablating conditions, and increasing the laser power from 5 W to 10 W mostly resulted in an increase in depth from 270 μm to 400 μm at a diameter of 450 μm (Figure 2C). By contrast, defocusing the substrate from the focal plane widened the microwell diameter. For instance, the diameter measurements for M_−3@15_ and M_−6@25_ were 700 μm and 1200 μm, respectively, and both were approximately 350 μm deep. These results could serve as guidelines for further optimization of microwells because, depending on the parameter used, they can be flexibly changed.

### 3.2. Generation of MCA

The formation of MCAs in different parametric microwells was examined at different days (Appendix A). Cells gradually formed MCAs in 1 day; these MCAs expanded in size on days 1, 3 and 4. The MCA size was confined by the size of the microwells, as indicated by the comparison of MCAs from the M_0@5_ and M_0@10_ microwells. MCAs in microwells formed strong cell–cell contacts without the boundary of individual cells obvious in an examination of the morphology. The viability and size generated from MCAs in different microwell sizes were evaluated (Figure 3). CMFDA live cell staining and Hoechst nuclear staining indicated good cell viability and revealed the shape of the MCAs harvested from M_0@5_, M_0@10_ and M_−3@15_ (Figure 3A). M_0@5_ and M_0@10_ exhibited uniform and spherical MCAs, with the average Feret diameter consistently measured between 250 μm and 280 μm (Figure 3B). By contrast, M_−3@15_ had round and smooth MCAs that were larger in size at ~520 μm with greater size variation than those from other microwells.

### 3.3. 2D and MCA High-Throughput Anticancer Drug Screening

To demonstrate that these easily fabricated microwells can be integrated with the conventional high-throughput workflow, a standard 96-well multi-well plate was used for preliminary drug screening verification (Figure 4). The morphology of 2D cultured cells and MCAs after treatment with 2 µM DOX was examined in 2D cultured cells and MCAs by SEM (Figure 4A). In the 2D control condition, Huh-7 cells were found to smoothly spread on cell culture substrate, revealing a standard 2D morphology with distinct lamellipodia and flat sheet-like structure (indicated by white arrows). However, under DOX treatment, cells showed a wrinkled morphology (pointed by white arrows), indicating a detrimental effect on cell growth and attachment. In the 3D condition, the MCAs had formed a cellular sphere with a clear indication of multiple individual cells. In the presence of DOX, impairment to the cells was observed in the form of a considerable number of cells detaching from the MCAs, leaving a partially hollow structure. 2D and MCAs generated by M_0@10_ were further assessed to evaluate the effects of DOX treatment on cell viability (Figure 4B). Dose-dependent responses were investigated for both conditions at concentrations ranging from 0 to 320 µM, and the IC_50_ values were 9.3 and 42.8 for the 2D and 3D MCAs (M_0@10_), respectively.

### 3.4. The Binding Capacity of ConA-SCHSs and MCAs

To investigate alternative therapeutic interventions against MCAs, other than the conventional drug screening approach, the photothermal effect was utilized by binding ConA-SCHSs with MCAs (Figure 5). As illustrated, ConA was first conjugated with SCHSs prior to application to MCAs (Figure 5A). The diameter of a SCHS was approximately 300 nm, and each shell was approximately 20 nm thick [20]. The SEM image showed that the SCHSs had a uniform size and shape. To demonstrate the specific binding between SCHSs and MCAs through conjugation of the ConA with the SCHSs, a preliminary specificity investigation was performed by incubating either SCHSs or ConA-SCHSs with MCAs for two hours (Figure 5B). The results showed that SCHSs alone were unable to attach to MCAs, as indicated by a smooth surface; in contrast, ConA-SCHSs were found to be present on the surface of MCAs, indicating increased interaction between MCAs and SCHSs through ConA. ConA lectin was previously found to attach to glycoproteins on the cell surface, such as mannose and glucose [21,22], and was previously reported to be a potential targeting agent, with an action based on cellular phagocytosis, for anticancer therapy [23].

Considering the ample presence of glycan on Huh-7 cells, the ConA binding capacity of SCHSs was further evaluated at different concentrations by measuring the fluorescence presented on MCAs (Figure 6). To find the optimal interaction through the fluorescence response, ConA-FITC was utilized to form ConA-FITC-SCHSs. We tested different proportions of ConA-FITC to SCHS from 0:0 to 500:200 µg/mL (Figure 6A). The results showed that the fluorescence intensity was proportional to the amount of ConA-FITC-SCHSs added, suggesting that the higher concentration corresponded to higher binding capacity. Among different concentration combinations, 500:200 µg/mL ConA-FITC:SCHS displayed the highest binding capacity of the SCHSs. We further examined whether a cytotoxic effect was induced at the elevated ConA:SCHS concentration under the same incubation time of 1 h using LIVE/DEAD stain (Appendix A). Three ConA-FITC:SCHS ratio combinations, 250:100, 500:200 and 750:300 µg/mL, were examined, and the results showed that the higher concentrations were relatively more detrimental to the survival of the MCAs. Therefore, the ConA-FITC:SCHS concentration ratio was determined as 500:200 µg/mL based on the synergistic results of a higher binding efficiency of the ConA-FITC-SCHSs and survival rate of the MCAs. The time required for each ConA-FITC-SCHSs to bind to the MCAs was investigated for between 0.5 and 2.5 h to determine the applicable incubation time for the interaction of the ConA-FITC-SCHCs with the MCAs (Figure 6B). The results indicated that at least 1.5 h was required for a favorable interaction between the ConA-FITC-SCHS and the MCAs to transpire. As a result, considering that prolonged incubation time may impose adverse effects, an intermediate time of 2 h was chosen for the following experiments.

### 3.5. Photothermal Treatment Through Bound ConA-SCHSs.

Photothermal treatment was further implemented by applying a NIR laser to ConA-SCHSs bound to MCAs (Figure 7). LIVE/DEAD cell staining of MCAs showed that the photothermal effect could induce cancer cell death under exposure to a 3 W/cm^2^ NIR laser for 20 min, which was indicated by an increase in red fluorescence. The same illumination condition was also applied to the control, but no obvious cell death was observed. The relative fluorescence ratio showed an eight-fold reduction in cell viability after photothermal treatment, indicating the preliminary therapeutic effect for cancer treatment using NIR laser radiation through ConA-SCHSs. In fact, photothermal therapy takes advantage of electromagnetic radiation for the treatment of various medical conditions, including cancer, by providing an alternative to traditional chemotherapy such as that based on DOX. Performing photothermal treatment is relatively easy and can yield fast healing with few complications [23]. Our results further suggested that photothermal therapy can be readily applied to the microwell platform. In addition, the ConA-SCHSs used in this study constituted a photothermal agent that demonstrated high specificity to Huh-7 cells and could be excited under NIR laser irradiation [20,24,25]. Despite the fact that the LIVE/DEAD assays showed that some cancer cells remained alive after treatment, the increase in the percentage of dead cells indicated the usefulness of this technique. Further research on the design and use parameters based on the present study may improve the applicability of this MCA-based photothermal treatment.

## 4. Discussion

Commercial CO_2_ lasers have served as the most common and reliable sources of laser beams since their introduction to the industry in 1965 [26], and they have gradually been applied to various medical and biomedical practices [27,28,29,30]. The application of single-point CO_2_ laser ablation on plane substrates has enabled rapid prototyping of microwells, as we and others have reported, using polyester [31], PDMS [32], PS and PMMA [18]. Among the different choice of materials, PS for microwells not only showed the smoothest surface profile but also the geometry most conducive for growing MCAs with the benefit of easy real-time optical observation in situ through standard inverted microscopy. In addition, PS is the most commonly used material for in vitro cell-based research. Therefore, the integration of PS microwells with conventional cell culture plasticware is potentially acceptable in most laboratory settings because it minimizes the need for more material. Existing PS cell cultures also facilitate integration with different dish formats for various purposes. We examined the laser ablation effects at different focal planes, confirming that wider microwells can be rapidly fabricated by tuning the defocused length from −3 mm to −6 mm. For large quantities of MCAs, an array of thousands of microwells can be readily fabricated in either a 60- or 100-mm untreated cell culture dishes. In contrast, decreasing the surface area to 12- or 96-well plates may be useful for cultures that require isolated conditions, thus facilitating the integration in the standard high-throughput screening (HTS) workflow.

It is generally presumed that the ability to form uniform MCAs indicates easily facilitated subsequent functional assays, such as those for migration [10], invasion [33] or drug screening [34]. Our results demonstrated that MCAs can be reliably grown at controllable sizes in microwells by increasing the power parameter of the laser ablation. The sizes of MCAs can be significantly fine-tuned to a few tenths of a micrometer in M_0@5_ and M_0@10_ of ~250 μm. Although M_−3@15_ can generate large MCAs of ~500 μm, the size distribution was less desirable. This result could be explained by the fact that a medium exchange step was usually performed 10 min after cell seeding to prevent uncontrolled aggregation of cells not attached to M_0@5_ and M_0@10_ [18,35]. However, given that M_−3@15_ exhibited a larger aspect ratio, which tends to disrupt cells during medium exchange operations, only a portion of the medium was exchanged, resulting in cells remaining in the gap of the microwell that may promote uneven MCA distribution. Recent reports demonstrated that controlling the density of the microwells by reducing the gaps between them can enhance the formation of regular uniform MCAs and save the precious sample from being depleted during the medium exchange procedure [15,36]. Although M_−3@15_ was not selected for subsequent functional assays owing to the issue presented, this outcome highlights the potential improvements that could be further implemented to empower the applicability of the presented method.

Our preliminary data suggested that 3D MCAs showed higher drug resistance than 2D monolayer cells, likely due to different drug interactions caused by the structural differences of the two distinct conditions. That is, in the 2D condition, cells are directly exposed to the drug during incubation; by contrast, the densely packed cell–cell junctions in MCAs piled into multiple layers and reduced the penetration of toxicants, which explains why only cells located at the periphery appeared to be damaged (Figure 4A). In fact, this resemblance of tissue structures was found to increase the resistance of anticancer drugs in 3D MCAs, as reported elsewhere [15]. Similarly, Breslin et al. used the HER2-positive breast cancer cell line as a 3D MCA model (BT474, HCC1954 and EFM192A) and found that cells cultured by the 3D method showed more drug resistance to anticancer agents than cells cultured in 2D conditions [37]. Although it is now generally accepted that 3D MCA models could better represent an avascular tumor model in vitro, there is, unlike for 2D screening, no drug screening standards for 3D models to date; e.g., the number of cells, choice of suspension culture or level of embedment into the ECM has not been finalized. As pointed out by Gong et al., the cytotoxicity test of the MCF-7 3D cell model showed that drug resistance increases with the number of cells [38]. In addition, Liu et al. showed that different sizes of HepG2 MCAs had different levels of drug resistance [39]. Even among the commercial or research-based microwell platforms, there has been a wide range of discrepancies regarding the choice of incubation time prior to drug administration, type of microwell substrate and number of MCAs used for each condition, thus prompting the call for an industrial standard that responds to and properly addresses a multi-platform and multi-cell line comparisons. Moreover, although it might be difficult for individual laboratories to meet industrial standards, the features described above highlight the endeavors to develop a simple, economical and reproducible microwell platform that could substantially advance the discovery of better drugs worldwide.

The importance of 3D cell culture in cancer research has attracted much attention because it enables cellular responses to take place in conditions that more closely resemble conditions in vivo. The 3D MCA model provides a pathophysiological microenvironment that is similar to the avascular tumor in vivo [40,41,42]. In this study, we reported a simple microwell that can be integrated with conventional cell culture plasticware in the generation of uniform-sized MCAs for drug screening and photothermal treatment. The microwells were produced by single-point CO_2_ laser ablation, and by adjusting the laser power and the focal plane, arrays of microwells with desirable sizes for forming MCAs can be conveniently produced. Compared to various in-house techniques or platforms using either microfabrication or laser, of which most require additional steps associated with material preparation and replica molding [38,43,44], the new alternative offered in this study can be combined with PS cell culture plasticware to extend the use of accessible material to suit different cell culture conditions. In addition, users can readily customize the location, size and arrangement of the microwells based on their experimental needs, greatly enabling user design flexibility and integration. 

Despite the many advantages of our present approach, some limitations should be noted for future reference. First, the large MCAs generated in our current platform (M_-3@15_) were undesirable primarily because the additional medium exchange step was not optimized due to the low aspect ratio of the microwells. This issue could be addressed by reducing the fine gaps between microwells to prevent unwanted cell lodging. However, given that the microwell also has a symmetric wing-like structure that protrudes at the edges after laser ablation, it is unclear whether the microwells are manufactured in a compactly arranged manner. Second, microwells after thermal ablation may present unwanted autofluorescence in the PS materials that are suitable only for bright-field observation during MCA culture. Further fluorescence-related investigation requires transferring the sample to a different container, such as a microtiter plate, dish or microscope slide. Third, although each microwell can be fabricated in less than 100 milliseconds, the amount of time could accumulate when a considerable number of microwells are needed. Last but not least, uneven fluorescence distribution was observed in both CMFDA and live/dead staining that were accounted primarily for preventing undesired MCA damage and loss during incubation of the staining reagents. As these steps were adopted from the 2D condition, optimization should be highlighted for future protocol improvement. Nevertheless, considering the wide range of fabrication flexibility along with minimal new material needed, it is thought that this technique can offer a simple alternative for most laboratories.

## 5. Conclusions

This study presented a simple and economical in-house microwell by direct laser ablation of conventional cell culture plasticware. The size of the microwells could be effectively controlled by adjusting the power and focusing parameters of the CO_2_ laser. The microwell platform enabled the formation of uniformly controllable sizes of MCAs in M_0@5_ and M0_@10_ with good cell viability. The integration of microwells with a 96-well plate affirmed a potential workflow for high-throughput 3D drug screening. The protocol using ConA(-FITC)-SCHSs was established to visualize the optimal interaction and assess the effects of photothermal treatment on the MCAs. It is envisioned that the results of microwell fabrication and integration for MCA formation will serve not only as a guide for further parameter optimization but also as a practical and conveniently incorporated means of enhancing the conventional workflow for biological and medical studies, thereby serving as an effective tool for the discovery of therapeutic modalities for cancer treatment.

## Figures and Tables

**Figure 1 materials-12-03308-f001:**
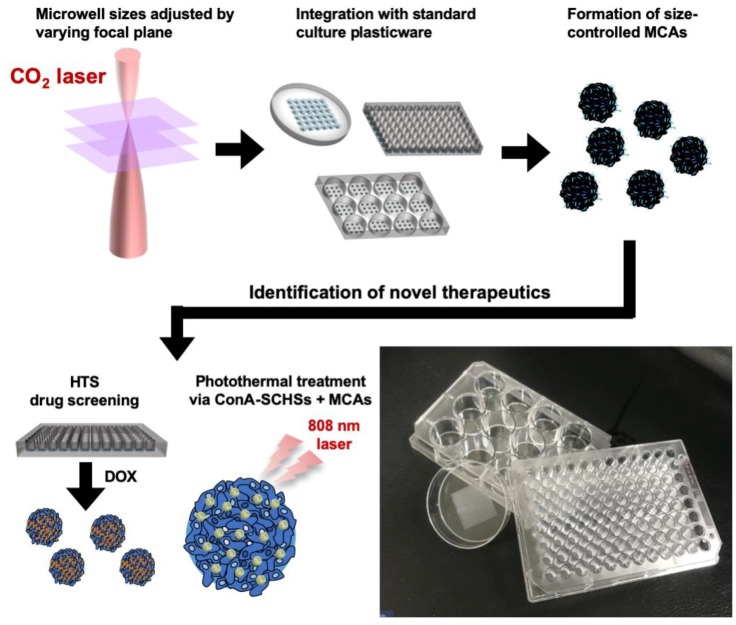
Schematic illustrations and an image of the rapid laser ablation of microwells integrated with standard culture plasticware for the identification of novel therapeutics through high-throughput screening (HTS) drug screening and photothermal treatment.

**Figure 2 materials-12-03308-f002:**
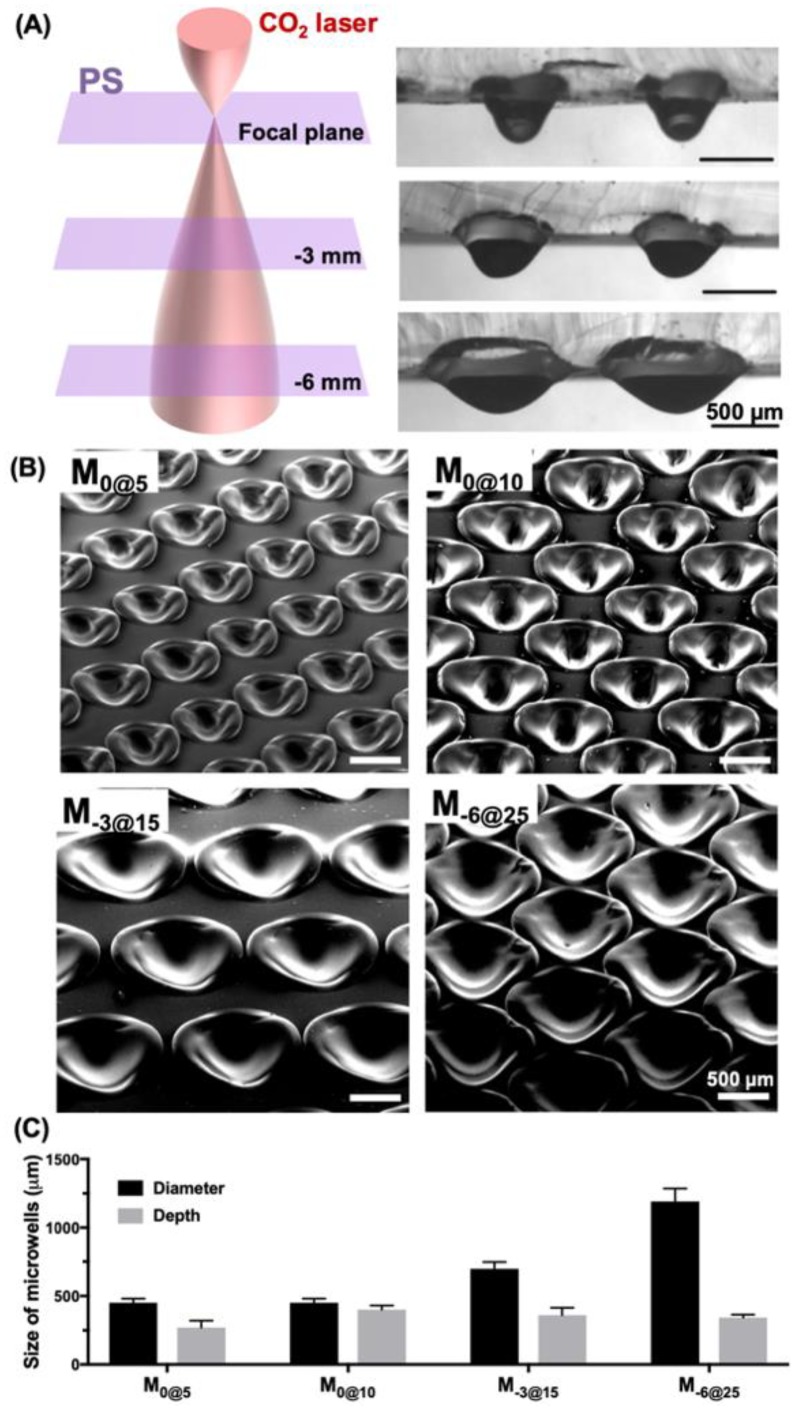
Microwells fabricated via in/out-of-focus laser ablation to generate a wide range of sizes. (**A**) Schematic illustration of incremental z-axis changes showing the gradually enlarged laser beam at different in/out-of-focus planes from 0 mm to −6 mm. (**B**) SEM images of the isometric view of microwells fabricated at different levels of laser power and various focusing planes. (**C**) Characterization of microwell diameters and depths at different levels of laser power and various focusing planes.

**Figure 3 materials-12-03308-f003:**
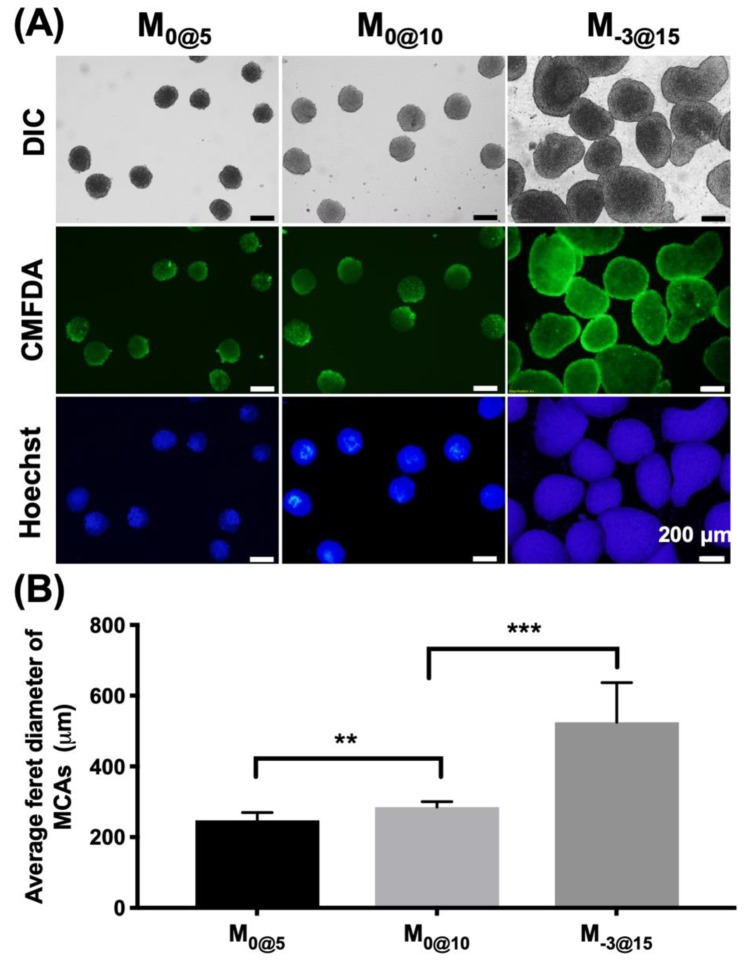
Comparison of multicellular cancer aggregates (MCAs) formed in different microwell conditions. (**A**) The images of the morphology of the MCAs formed in different microwells. (**B**) The average Feret diameter was measured for the MCAs formed in each type of microwell.

**Figure 4 materials-12-03308-f004:**
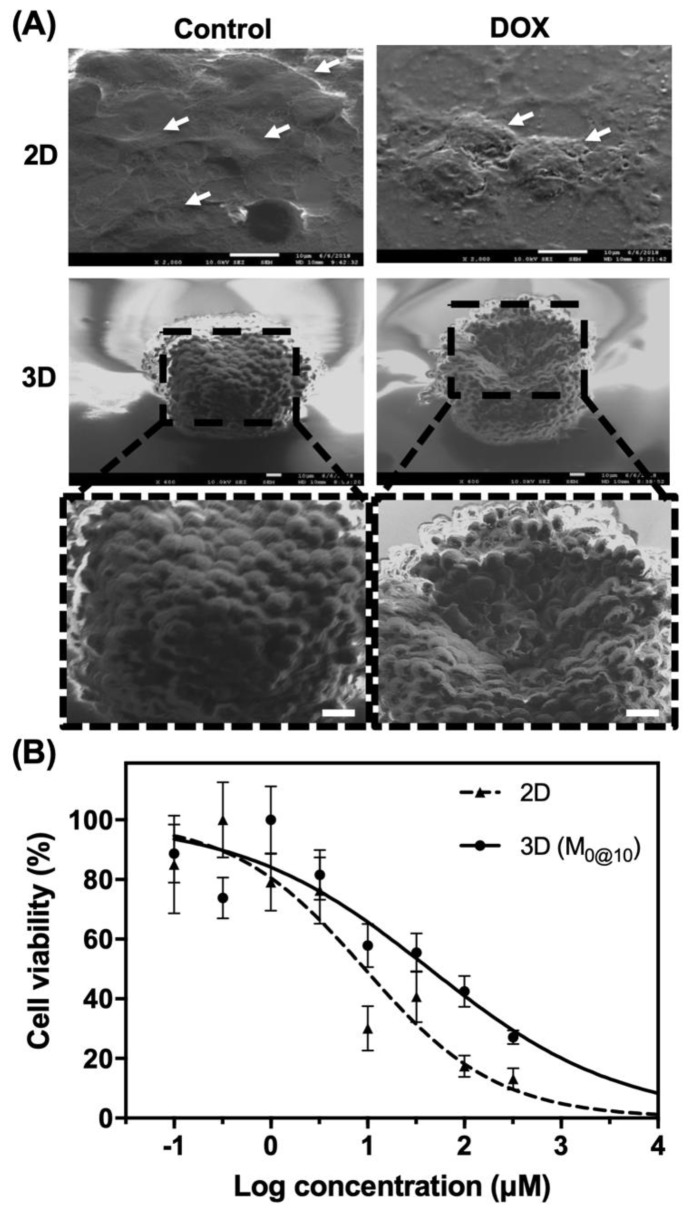
Application of DOX in the 2D condition and with MCAs. (**A**) SEM image of Huh-7 in 2D/3D culture, with/without DOX, cultured for 5 days. (**B**) Dose-response curve of cell viability after treatment with DOX. Scale bars: 10 μm.

**Figure 5 materials-12-03308-f005:**
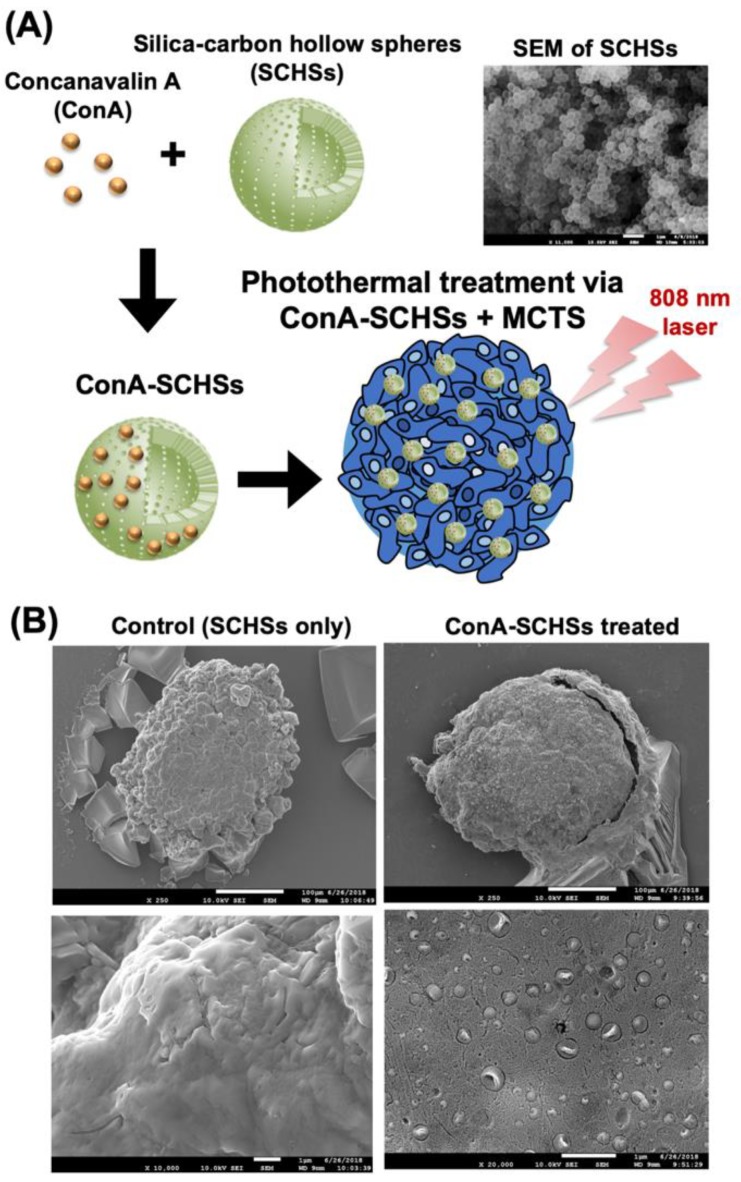
Illustration and SEM images of ConA-SCHSs (concanavalin A-silica–carbon hollow spheres) conjugation for the photothermal treatment of the MCAs. (**A**) ConA bound to uniformly sized SCHSs (shown in SEM) was applied to the MCAs and irradiated by an 808 nm laser for the photothermal treatment. (**B**) The SEM images of the MCAs treated with SCHSs or ConA-SCHSs.

**Figure 6 materials-12-03308-f006:**
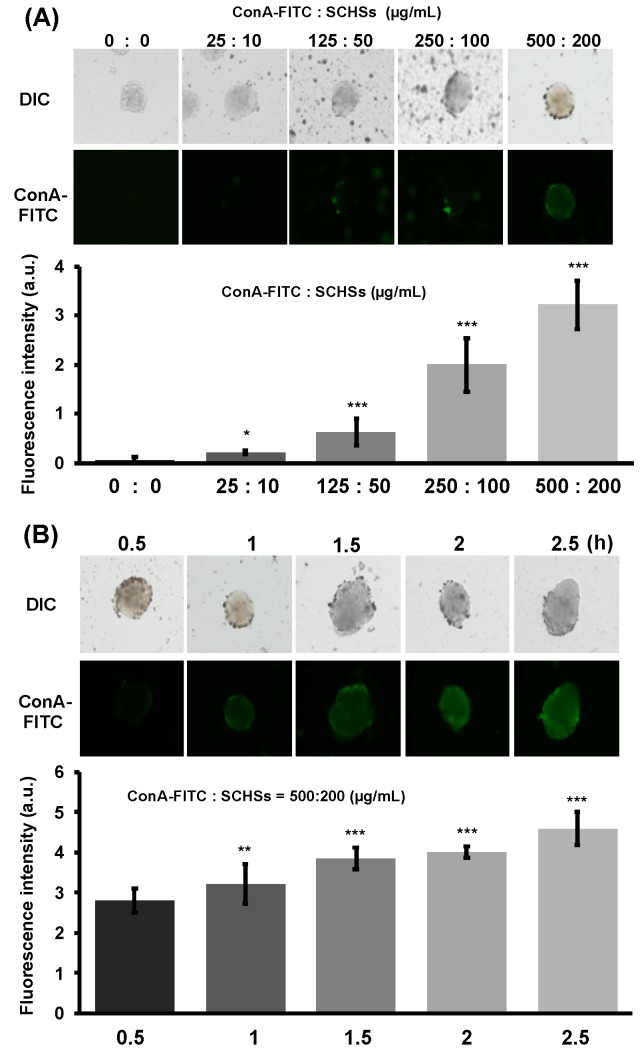
Evaluation of the proper proportion of conjugated ConA-FITC to SCHSs and the time it is exposed to MCAs. (**A**) FITC-labeled ConA (ConA-FITC) conjugated with SCHSs showing an increased binding ability to MCAs when the concentration was increased. (**B**) Time evaluated for binding of ConA-FITC-SCHSs to MCAs.

**Figure 7 materials-12-03308-f007:**
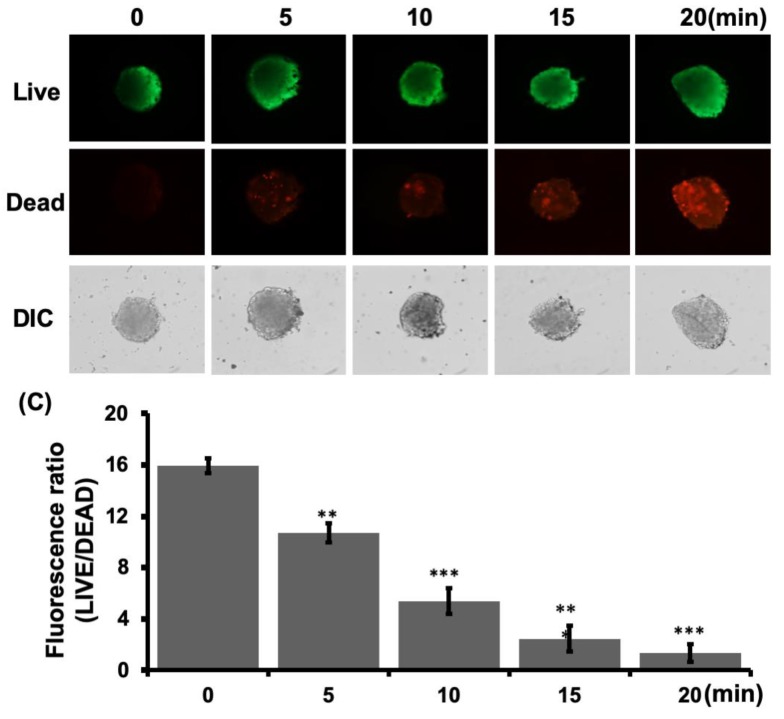
Photothermal treatment through bound ConA-SCHSs. (**A**) Live cell staining of MCAs showing increased red fluorescence of dead cells after photothermal treatment. (**B**) Evaluation of the relative fluorescence exhibiting the ratio of live to dead cells after staining cells in control and treated conditions.

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
