# Peer review of "Simple In-House Fabrication of Microwells for Generating Uniform Hepatic Multicellular Cancer Aggregates and Discovering Novel Therapeutics"

_materials, 2019, doi:10.3390/ma12203308_

Round 1
Reviewer 1 Report
A well written and well-presented study which offers a potentially cost-effective solution to the issue of growing consistent 3D cultures.
The technology looks very reproducible and easy to translate form one lab to another, with my only concern being the cost and availability of the laser engraver. Whilst this may be an affordable equipment item in a large multi-disciplinary institution, or labs where this technique is being used in high-throughput screening, but it may be beyond the reach of small research laboratories where smaller, 'one-off' experiments are carried out.
Author Response
Response to Reviewer 1:
Point 1: A well written and well-presented study which offers a potentially cost-effective solution to the issue of growing consistent 3D cultures.
The technology looks very reproducible and easy to translate form one lab to another, with my only concern being the cost and availability of the laser engraver. Whilst this may be an affordable equipment item in a large multi-disciplinary institution, or labs where this technique is being used in high-throughput screening, but it may be beyond the reach of small research laboratories where smaller, 'one-off' experiments are carried out.
We thank the reviewer’s comments. In fact, the price for different types laser sources can vary significantly. In this study, the choice of CO2 has served as the most common and reliable sources of laser since its introduction in 1965. Although the price of the CO2 laser is contingent to the desired power output required, a 10W CO2 laser suitable for the current application can be easily quoted under $12,000. This price is fairly affordable to most of the biological and medical laboratories, compared to most of the common lab equipment such as ultra-low temperature freezer, centrifuge, incubators and so forth. However, these potential benefits do not exclude the fact that, as rightly pointed out by the reviewer, small laboratories may only seek to perform few sets of experiments instead of a series of it. In this regard, the commercial products may offer some level of convenience. However, as also pointed out in the manuscript, commercially available products also face limitations, not only because of their costly nature but also because these microwells are produced to fixed settings such that they lack the flexibility for changing dimensions or arrangements.
Therefore, with these points being discussed, we envision that this technique will be able to reach a greater scientific community through the influence and open access convenience of the journal. We hope you will find our manuscript fit for acceptance and publication.

Reviewer 2 Report
The authors describe a new method for creating microwells in commercially available cell culture dishes and describe the possibilities and limitations in a really fair way. Even to readers not having access to laser imprinting facility, the impact of microwell size and structure to MCA/tumor spheroid cultures is of value. In the discussion the authors nicely point out the chances and limitations of 3D cultures for drug testing as of today. So several other publications show the merits of avascular 3D cultures, yet comparability is not reached so far.
Major points:
1. Since this paper is a method description on cell culture, the M&M section needs to be more detailed to really envision the advantages and improvements (see also under ‘minor comments’ for details).
2. A list of abbreviations would be helpful.
3. For the generation of MCA different microwell sizes were used. It is shown in Fig. 3 that smaller microwells lead to smaller MCA after 4 days. Looking at Fig. 2 the reviewer gets the impression, that MCA do not really gain in size between days 3 and 4. The more uniform MCA sizes could also indicate a growth arrest. Are there any data about growth curves? Especially could the state of MCA (still growing or in arrest due to size limitation) influence the differences of 2D vs 3D cultures in response to treatment? If 3D cultures are used as cancer treatment model, their state must be considered. To this reviewer it seems essential, that at least cells at the rim of MCA are proliferating (reflecting the viable rim of a tumor). In Figure 3 the fluorescent images are not really indicative of cell status. Hoechst could show intact vs apoptotic nuclei, yet the magnification here does not allow for that. It may be difficult to image individual cells/nuclei within a MCA by epifluorescence imaging. CMFDA staining shows a brighter rim, yet this could be due to dye penetration (and lower availability in the MCA core).
1. Regarding the images of DOX treated cells in Figure 4, the reviewer suggest to clarify/highlight the viable cells for 2D - and more explicitly describe the phenotype, since this is hard to identify for readers used to brightfield/fluorescent images (usually shown for cell viability in 2D).
2. Since the images in Figure 3 A may indicate a dye penetration issue for MCA, it would be nice to have clarification if for the viability determination upon treatment, the MCA have been dissolved or if it was assured, that cells in the MCA core were measured accordingly.
3. Figure 5 B needs more detailed and clarified description (especially in the legend and regarding the lower panel). Does ‘Control’ vs ‘ConA-SCHSs treated’ mean, unconjugated SCH without laser treatment vs ConA-SHCs with laser treatment? Please clarify this for the readers.
Minor points:
1. It would be helpful to point out if there is a difference between MCA and the commonly use term ‘spheroids’
2. Ref 1: date is missing (publication or download)
3. Line 46: prognosis (typing error)
4. Ref 2,3 and 4 are from 2014. That is 5 years old. It seems that there should have been some new developments in diagnosis and treatment (= more recently published review papers).
5. Line 124: what does ‘long time’ mean? Have there been any storage time evaluations/experiments?
6. Line 126: please specify what ‘Pluronics’ was used
7. Line 130: In petridishes a medium change was performed – how long after seeding the cells. Was a similar procedure performed for the well plates? How many cells were deposited per microwell? Was that quantified or controlled for?
8. Line 136: What ‘parameters’ were analyzed?
9. M&M 2.3 and 2.4: If the reviewer understands the methodology correctly, the microwells were generated in a well/petridish. Reagents were then put into this lager structure. Please describe this consistently for all 3 sizes (petri dish, 12 well and 96 well plates) or specify which procedures were done for which sizes.
10. Line 143: 200µL/well is not a ‘density’.
11. Line 142/143: Was cell morphology determined in floating MCA in a quite large volume or did they adhere (incubation time and possible)?
12. 145-148: When was the Live/dead staining performed (still in the microwells)?
13. Line 168: Was the seeded cell number the same in 2D and 3D? E. g. what volume was used for 2D? 200µL as for the microwells?
14. Line 241 and Figure S3: In text ‘timelapse’ imaging is stated. Yet the microwells in Figure S3 do not seem to be the same for the 3 timepoints. Is this due to (inherent) inaccuracies in plate positioning on the microscope or are this examples of typical MCA per timepoint (different ROI/microwells)? Please specify.
15. Figure 4: bar size is missing.
16. Figure S4 and 7: please comment on the uneven staining within the MCA shown.
17. Figure 6: Please clarify in line 317 the meaning of ‘at 2h’.
18. Line 332: How was high specificity of ConA to Huh7 cells determined?
Author Response
Response to Reviewer 2:
The authors describe a new method for creating microwells in commercially available cell culture dishes and describe the possibilities and limitations in a really fair way. Even to readers not having access to laser imprinting facility, the impact of microwell size and structure to MCA/tumor spheroid cultures is of value. In the discussion the authors nicely point out the chances and limitations of 3D cultures for drug testing as of today. So several other publications show the merits of avascular 3D cultures, yet comparability is not reached so far.
Major points:
Point 1: Since this paper is a method description on cell culture, the M&M section needs to be more detailed to really envision the advantages and improvements (see also under ‘minor comments’ for details).
We have rewritten the entire sections of 2.1 Microwells and 2.3. Formation of MCAs, as well as the majority of 2.4. MCA morphology and viability assessment. Detailed responses have been appended in the below answers.
Point 2: A list of abbreviations would be helpful.
We have added a list of abbreviations before the Introduction for ease of reference.
“Abbreviations: HCC, hepatocellular carcinoma; 2D, two-dimensional; 3D, three-dimensional; MCTs, multicellular tumor spheroids; MCAs, multicellular cancer aggregates; ECM, extracellular matrix; DOX, doxorubicin; ConA, concanavalin A; SCHSs, silica-carbon hollow spheres; CMFDA, 5-chloromethyl fluorescein diacetate; SEM, scanning electron microscopy; PS, polystyrene; PMMA, poly(methyl methacrylate).”
Point 3: For the generation of MCA different microwell sizes were used. It is shown in Fig. 3 that smaller microwells lead to smaller MCA after 4 days. Looking at Fig. 2 the reviewer gets the impression, that MCA do not really gain in size between days 3 and 4. The more uniform MCA sizes could also indicate a growth arrest. Are there any data about growth curves?
If we understand the reviewer’s question correctly, the main question lies in the difference of MCAs sizes generated by similar sizes of M0@5 and M0@10. We think this result was primarily due to the difference in aspect ratio of the two microwell parameters. We found that the smaller the aspect ratio, the easier for the cells being aspirated from the microwells during the medium exchange step. Unfortunately, we did not test the growth curve of the MCAS. We have added the detailed steps for medium exchange after cell seeding in the section “2.3 Formation of MCAs.”
Point 4: Especially could the state of MCA (still growing or in arrest due to size limitation) influence the differences of 2D vs 3D cultures in response to treatment?
Although steep oxygen gradients could be observed in MCAs above 200 mm, previous reports usually used MCAs larger than 500 mm for growth arrest studies, as well as prolonged culture time to at least 7 days and beyond [1,2]. While some confounding effects may be present, considering the size of the MCAs for drug screenings was generated at around 280 mm and combining the MCAs were studied on day 4 or 5 (including additional day for drug incubation), we attributed this outcome due to the resemblance of tissue structures that increased the resistance of anticancer drugs in 3D MCAs.
Riffle, S.; Pandey, R.N.; Albert, M.; Hegde, R.S. Linking hypoxia, DNA damage and proliferation in multicellular tumor spheroids. BMC Cancer 2017. Grimes, D.R.; Kelly, C.; Bloch, K.; Partridge, M. A method for estimating the oxygen consumption rate in multicellular tumour spheroids. J. R. Soc. Interface 2014.
Point 5: If 3D cultures are used as cancer treatment model, their state must be considered. To this reviewer it seems essential, that at least cells at the rim of MCA are proliferating (reflecting the viable rim of a tumor). In Figure 3 the fluorescent images are not really indicative of cell status. Hoechst could show intact vs apoptotic nuclei, yet the magnification here does not allow for that. It may be difficult to image individual cells/nuclei within a MCA by epifluorescence imaging. CMFDA staining shows a brighter rim, yet this could be due to dye penetration (and lower availability in the MCA core).
We thank the review’s comments. We think the gradient of the fluorescence signal shown in the figure was primarily due to the staining procedure of the CMFDA, where the MCAs were kept in the microwells to avoid excessive pipetting steps and only retrieved out prior to the optical and fluorescent investigation.
As shown in M&M in page 9 and Discussion in page 15.
“For ease of imaging, the MCAs were transferred to a 96-well plate at a volume of 200 µL/well in suspension, and their morphology was evaluated using an inverted fluorescence microscope immediately after the MCAs settled to the bottom of the plate. Photothermal treatment on MCAs were assessed using the LIVE/DEAD viability/cytotoxicity assay kit (Invitrogen, Carlsbad, CA, USA) to analyze cell viability. Calcein-AM staining solution (1:2,000 dilution) and EthD-1 staining solution (1:1,000 dilution) were added and incubated at 5% CO2 and 37°C for 1 h prior to image capture and examination. It should be noted that to avoid breakage or dislodging of cells from MCAs during the pipetting steps, all MCAs were kept in the microwells throughout the entire staining procedures, and only retrieved prior to the optical and fluorescent investigation.”
“Last but not least, uneven fluorescence distribution was observed in both CMFDA and live/dead staining that were accounted primarily for preventing undesired MCA damage and loss during incubation of the staining reagents. As these steps were adopted from the 2D condition, optimization should be highlighted for future protocol improvement.”
Point 6. Regarding the images of DOX treated cells in Figure 4, the reviewer suggest to clarify/highlight the viable cells for 2D - and more explicitly describe the phenotype, since this is hard to identify for readers used to brightfield/fluorescent images (usually shown for cell viability in 2D).
We have added the white arrows in Figure 4A to point out the morphology that we hoped to describe to the readers. We have also added an additional information depicting the difference between healthy and unhealthy morphology for 2D. The revision is reflected in page 9 of the manuscript.
“In the 2D control condition, Huh-7 cells were found to smoothly spread on cell culture substrate, revealing a standard 2D morphology with distinct lamellipodia and flat sheet-like structure (indicated by white arrows). However, under DOX treatment, cells showed a wrinkled morphology (pointed by white arrows), indicating a detrimental effect on cell growth and attachment.”
Point 7. Since the images in Figure 3 A may indicate a dye penetration issue for MCA, it would be nice to have clarification if for the viability determination upon treatment, the MCA have been dissolved or if it was assured, that cells in the MCA core were measured accordingly.
As explained in the previous answer, although the staining was slightly uneven due the staining procedure, the distinct fluorescent signal indicated MCAs were in a homogeneous condition of both the size and viability.
Point 8. Figure 5 B needs more detailed and clarified description (especially in the legend and regarding the lower panel). Does ‘Control’ vs ‘ConA-SCHSs treated’ mean, unconjugated SCH without laser treatment vs ConA-SHCs with laser treatment? Please clarify this for the readers.
Yes, control group represented the SCHSs incubated with MCAs only. We have added “Control (SCHSs only)” in the figure 5.
Minor points:
Point 1: It would be helpful to point out if there is a difference between MCA and the commonly use term ‘spheroids’
First of all, we thank the reviewer for pointing out this term clarification issue. The idea of using MCA was to defined it based on its formation process, where cells aggregated into a clump of cells over a period of time. Although a review written by Weiswald et al. [3] summarized that the term “aggregates” has been primarily used to described loose packages of cells, we also noticed the usage remained inconsistent over different publications [4,5]. To ensure the present study can be best reached by the relevant scientific community, we have adopted the term “multicellular tumor spheroids” in the keywords and the introduction.
Weiswald, L.-B.; Bellet, D.; Dangles-Marie, V. Spherical Cancer Models in Tumor Biology. Neoplasia 2015, 17, 1–15. Klymenko, Y.; Johnson, J.; Bos, B.; Lombard, R.; Campbell, L.; Loughran, E.; Stack, M.S. Heterogeneous Cadherin Expression and Multicellular Aggregate Dynamics in Ovarian Cancer Dissemination. Neoplasia (United States) 2017. Klymenko, Y.; Kim, O.; Loughran, E.; Yang, J.; Lombard, R.; Alber, M.; Stack, M.S. Cadherin composition and multicellular aggregate invasion in organotypic models of epithelial ovarian cancer intraperitoneal metastasis. Oncogene 2017.
Point 2: Ref 1: date is missing (publication or download)
We have corrected the source and the year of the Ref 1 accordingly.
“Ferlay, J.; Soerjomataram, I.; Ervik, M.; Dikshit, R.; Eser, S.; Mathers, C. Globocan 2012: Cancer Incidence and Mortality Worldwide 2012 Cancer Fact Sheet; 2013;”
Point 3: Line 46: prognosis (typing error)
We have corrected the typo accordingly.
“Hepatocellular carcinoma (HCC) is the most common type of primary liver cancer and is associated with poor prognosis due to …”
Point 4: Ref 2,3 and 4 are from 2014. That is 5 years old. It seems that there should have been some new developments in diagnosis and treatment (= more recently published review papers).
We have recited the Ref 2, 3 and 4 using more recent literatures.
“2. Bertuccio, P.; Turati, F.; Carioli, G.; Rodriguez, T.; La Vecchia, C.; Malvezzi, M.; Negri, E. Global trends and predictions in hepatocellular carcinoma mortality. J. Hepatol. 2017.
Llovet, J.M.; Montal, R.; Sia, D.; Finn, R.S. Molecular therapies and precision medicine for hepatocellular carcinoma. Nat. Rev. Clin. Oncol. 2018. Nault, J.C.; Galle, P.R.; Marquardt, J.U. The role of molecular enrichment on future therapies in hepatocellular carcinoma. J. Hepatol. 2018.”
Point 5: Line 124: what does ‘long time’ mean? Have there been any storage time evaluations/experiments?
In fact, we did not check the exact storage condition and time over a prolonged period. Internally, we have used the fabricated plates up to three months old without encountering issues. Therefore, we have revised the statement that encompasses our internal validation accordingly in page 4.
“The fabricated microwells could be stored on the shelf for up to three months…”
Point 6: Line 126: please specify what ‘Pluronics’ was used
We have added the complete product information.
“pluronic (F127, Sigma–Aldrich, St Louis, MO, USA)”
Point 7: Line 130: In petridishes a medium change was performed – how long after seeding the cells. Was a similar procedure performed for the well plates? How many cells were deposited per microwell? Was that quantified or controlled for?
We have expanded the method section to encompass the entire medium exchange procedure. Medium was exchanged after 10 mins of cell seeding. Each microwell contained roughly around 100 cells.
“In the petri dishes and 12-well plate, excessive cells that did not lodge into the microwells were removed through medium exchange at the edge of cell culture surface after 10 min of cells seeding in room temperature [19].”
Point 8. Line 136: What ‘parameters’ were analyzed?
We have made the correction and remove the term “parameters”
“The morphology of the MCAs generated in microwells was assessed by first removing the culture media.”
M&M 2.3 and 2.4: If the reviewer understands the methodology correctly, the microwells were generated in a well/petridish. Reagents were then put into this lager structure. Please describe this consistently for all 3 sizes (petri dish, 12 well and 96 well plates) or specify which procedures were done for which sizes.
We have rewritten the sections as shown below.
“The fabricated microwells could be stored on the shelf for up to three months and washed twice with 75% ethanol to remove debris from the laser fabrication and for disinfection prior to use. Before cell seeding, the microwells were immersed in 0.2% pluronic (F127, Sigma–Aldrich, St Louis, MO, USA) for 30 min to prevent undesired cell attachment to the microwell plate, followed by two phosphate-buffered saline (PBS; Invitrogen, Carlsbad, CA, USA) washes. Then, a cell concentration of 1 × 105 cells/mL was determined based on the surface area of the culture plasticwares and seeded in the microwell-modified 60 mm petri dishes (5 mL of cell suspension media/dish), and 0.2 × 105 cells/mL and 0.375 × 105 cells/mL for 12- and 96-well plates (1 mL and 200 µL of cell suspension media/well), respectively. In the petri dishes and 12-well plate, excessive cells that did not lodge into the microwells were removed through medium exchange at the edge of cell culture surface after 10 min of cells seeding in room temperature [19]. In 96-well plate, due to the limited surface area difficult for exchanging the medium without unaffecting the cells, only half of the medium was replaced daily. The formation of MCAs was achieved by incubating cells under the same cell culture conditions for 4 days. The MCAs were retrieved by the pipetting culture medium several times to flush the MCAs out of the microwells.”
Point 10. Line 143: 200µL/well is not a ‘density’.
We have changed the word “density” to “volume”.
“For ease of imaging, the MCAs were transferred to a 96-well plate at a volume of 200 µL/well in suspension…”
Point 11. Line 142/143: Was cell morphology determined in floating MCA in a quite large volume or did they adhere (incubation time and possible)?
The MCAs were imaged immediately after it settled to the bottom of the new 96-well plate. The clarifications have been made in the manuscript.
“For ease of imaging, the MCAs were transferred to a 96-well plate at a volume of 200 µL/well in suspension, and their morphology was evaluated using an inverted fluorescence microscope immediately after the MCAs settled to the bottom of the plate.”
Point 12. 145-148: When was the Live/dead staining performed (still in the microwells)?
Yes, the live/dead staining was performed while the MCAs remained in the microwells.
Point 13. Line 168: Was the seeded cell number the same in 2D and 3D? E. g. what volume was used for 2D? 200µL as for the microwells?
We thank the reviewer for pointing out this erratum. The corrected cell seeding density has been used.
“For the 2D condition, Huh-7 cells were cultured in a treated 96-well plate (0.375 x 105 cells/mL).”
Point 14. Line 241 and Figure S3: In text ‘timelapse’ imaging is stated. Yet the microwells in Figure S3 do not seem to be the same for the 3 timepoints. Is this due to (inherent) inaccuracies in plate positioning on the microscope or are this examples of typical MCA per timepoint (different ROI/microwells)? Please specify.
We thank the reviewer for pointing out this erratum. Figure S3 (has become Figure S4 due to addition of Figure S1) was showing cells located at different microwells over time. We have corrected the term from the manuscript.
“The formation of MCAs in different parametric microwells was examined at different days (Figure S4).”
Point 15. Figure 4: bar size is missing.
Scale bars were drawn to reflect 10 mm in length. Since all 6 figures in Figure 4A labeled with the same dimension, we have added this information in the legend instead of putting it into the figure.
“Figure 4. Application of DOX in the 2D condition and with MCAs. (A) SEM image of Huh-7 in 2D/3D culture, with/without DOX, cultured for 5 days. (B) Dose-response curve of cell viability after treatment with DOX. Scale bars: 10 mm.”
Point 16. Figure S4 and 7: please comment on the uneven staining within the MCA shown.
We think the gradient of the fluorescence signal shown in the figure was primarily due to the staining procedure of the live/dead, where the MCAs were kept in the microwells to avoid excessive pipetting steps and only retrieved out prior to the optical and fluorescent investigation.
As shown in M&M in page 9 and Discussion in page 15.
“For ease of imaging, the MCAs were transferred to a 96-well plate at a volume of 200 µL/well in suspension, and their morphology was evaluated using an inverted fluorescence microscope immediately after the MCAs settled to the bottom of the plate. Photothermal treatment on MCAs were assessed using the LIVE/DEAD viability/cytotoxicity assay kit (Invitrogen, Carlsbad, CA, USA) to analyze cell viability. Calcein-AM staining solution (1:2,000 dilution) and EthD-1 staining solution (1:1,000 dilution) were added and incubated at 5% CO2 and 37°C for 1 h prior to image capture and examination. It should be noted that to avoid breakage or dislodging of cells from MCAs during the pipetting steps, all MCAs were kept in the microwells throughout the entire staining procedures, and only retrieved prior to the optical and fluorescent investigation.”
“Last but not least, uneven fluorescence distribution was observed in both CMFDA and live/dead staining that were accounted primarily for preventing undesired MCA damage and loss during incubation of the staining reagents. As these steps were adopted from the 2D condition, optimization should be highlighted for future protocol improvement.”
Point 17. Figure 6: Please clarify in line 317 the meaning of ‘at 2h’.
Thanks for pointing out this erratum. Since the legend was depicting the time evaluated for binding of ConA-FITC-SCHSs to MCAs, “at 2h” was removed from the paragraph.
“…(B) Time evaluated for binding of ConA-FITC-SCHSs to MCAs.”
Point 18. Line 332: How was high specificity of ConA to Huh7 cells determined?
The high specificity of ConA-SCHSs to Huh-7 cells was primarily evaluated in our previous studies. We have added the references [20] and [25] accordingly for clarifications.
“20. Chen, Y.-C.; Chiu, W.-T.; Chen, J.-C.; Chang, C.-S.; Hui-Ching Wang, L.; Lin, H.-P.; Chang, H.-C. The photothermal effect of silica–carbon hollow sphere–concanavalin A on liver cancer cells. J. Mater. Chem. B 2015, 3, 2447–2454.
Chen, Y.C.; Chiu, W.T.; Chang, C.; Wu, P.C.; Tu, T.Y.; Lin, H.P.; Chang, H.C. Chemo-photothermal effects of doxorubicin/silica-carbon hollow spheres on liver cancer. RSC Adv. 2018.”
